# Short-duration splice promoting compound enables a tunable mouse model of spinal muscular atrophy

Anne Rietz[1], Kevin J Hodgetts[2], Hrvoje Lusic[2], Kevin M Quist[1], Erkan Y Osman[3] (iD), Christian L Lorson[3] (iD), Elliot J Androphy[1] (iD)

**Spinal muscular atrophy (SMA) is a motor neuron disease and the leading genetic cause of infant mortality. SMA results from insufficient survival motor neuron (SMN) protein due to alternative splicing. Antisense oligonucleotides, gene therapy and splicing modifiers recently received FDA approval. Although severe SMA transgenic mouse models have been beneficial for testing therapeutic efficacy, models mimicking milder cases that manifest post-infancy have proven challenging to develop. We established a titratable model of mild and moderate SMA using the splicing compound NVS-SM2. Administration for 30 d prevented development of the SMA phenotype in severe SMA mice, which typically show rapid weakness and succumb by postnatal day 11. Furthermore, administration at day eight resulted in phenotypic recovery. Remarkably, acute dosing limited to the first 3 d of life significantly enhanced survival in two severe SMA mice models, easing the burden on neonates and demonstrating the compound as suitable for evaluation of follow-on therapies without potential drug–drug interactions. This pharmacologically tunable SMA model represents a useful tool to investigate cellular and molecular pathogenesis at different stages of disease.**

## Introduction

Spinal muscular atrophy (SMA) afflicts ~1 in 6,000–10,000 live births, and half succumb within 2 yr (Verhaart et al, 2017). SMA results from insufficient survival motor neuron (SMN) protein. The *SMN1* gene, located on human chromosome 5q13.2, is duplicated, resulting in the nearly identical *SMN2* gene possessing a nucleotide transition (C → T) in exon 7, causing exon skipping and loss of the terminal 17 amino acids of the SMN protein (Lefebvre et al, 1995; Lorson et al, 1999; Monani et al, 1999). These alternatively spliced *SMN2* transcripts yield a highly unstable protein, SMNΔ7 (Lorson & Androphy, 2000). Only 10–15% of *SMN2* mRNAs produce full-length functional SMN protein.

SPINRAZA (nusinersen), an antisense oligonucleotide, ZOLGENSMA (onasemnogene abeparvovec-xioi), an AAV-9 based gene

therapy, and Risdiplam, a splicing molecule, have recently been FDA-approved for SMA; SPINRAZA and Risdiplam for all forms of SMA, and ZOLGENSMA for children under 2 yr. The other *SMN2* splicing modifier, Branaplam, is currently in Phase 2 for type I (NCT02268552). In SMA type I, clinical trial data indicate reduced lethality and achievement of important motor milestones following intervention with the three FDA-approved drugs. Motor functions stabilized in SMA type II patients instead of slowly declining. Risdiplam improved the Gross Motor Function Measure scale in SMA type II/III children aged 2 yr and older compared with placebo control (Dangouloff & Servais, 2019). Nonetheless, some patients did not respond to treatment, and there is a strong inverse correlation between the age at which treatment began and efficacy (Dangouloff & Servais, 2019). This highlights the need for co-therapy investigation, as one SMN-modifying agent may not be sufficient to completely improve motor skills and disease severity.

The SMNΔ7 SMA (FVB.Cg-Tg(SMN2*delta7) 4299AhmbTg(SMN2) 89Ahmb *Smn1tm1Msd*/J)) mouse model is most commonly used for testing SMA therapeutics. These mice lack murine *Smn* and express an intact human *SMN2* gene plus SMNΔ7 cDNA (Le et al, 2005). SMNΔ7 mice develop a severe SMA phenotype with impaired motor function and low body weight with an average life span of 12–13 d (Le et al, 2005). The SMNΔ7 mouse breeding scheme produces a predicted 25% litter with the SMA genotype. The less-used, slightly more severe "Li" or "Taiwanese" SMA mouse model (Jackson Labs; FVB.Cg-Smn1tm1HungTg(SMN2)2Hung/J.) also lacks murine *Smn* and expresses the human *SMN2* transgene (Hsieh-Li et al, 2000). These mice display low body weight, gastrointestinal dysfunction, and succumb by postnatal day (PND) 11 (Hsieh-Li et al, 2000; Sintusek et al, 2016). Their breeding scheme results in 50% of the litter developing the SMA-like phenotype. After disease progression, both mouse models exhibit necrosis of the ears, tail, and digits because of vascular thrombosis. Similarly, digital necrosis has been reported in infants with severe SMA (Araujo et al, 2009; Rudnik-Schoneborn et al, 2010). Both mouse models have marked reduction in the spleen size (Khairallah et al, 2017), which is recapitulated in the less severe *Smn*[2B/-] mouse model (Khairallah et al, 2017) that expresses a knock-in mutation disrupting splicing of endogenous

[1]Department of Dermatology, Indiana University School of Medicine, Indianapolis, IN, USA    [2]Laboratory for Drug Discovery in Neurodegeneration, Brigham and Women's Hospital, Harvard Medical School, Cambridge, MA, USA    [3]Department of Veterinary Pathobiology, Bond Life Sciences Center, College of Veterinary Medicine, University of Missouri, Columbia, MO, USA

Correspondence: eandro@iu.edu

*Smn* and survives ~1 mo (Hammond et al, 2010; Sleigh et al, 2011; Bowerman et al, 2012; Quinlan et al, 2019). The C+/+ mouse model (Jackson Lab; FVB.129(B6)-*Smn1tm5*(*Smn1/SMN2*)*Mrph*/J) is the mildest genetic model of SMA and exhibits low body weight with very mild impairment of a subset of motor functions, but has a normal life span (Osborne et al, 2012). This model has been used to investigate in vivo activity of small molecules. These transgenic SMA models are well-characterized and are the go-to standard for therapeutic testing. With the current advances of SMA treatment options, there is now a need to study co-therapies as well as models that more resemble SMA types II and III. To address this need, research has focused on non-genetic approaches with motor dysfunction beginning later in life.

Non-genetic mild SMA mouse models are typically generated in SMNΔ7 or *Smn²ᴮ* mice, although a small number of studies use the 5058 model. Daily administration of *SMN2* splicing modifier SMN-C3 at a suboptimal dose in SMNΔ7 mice induces a milder SMA phenotype (Feng et al, 2016) with low body weight and a median life span of 28 d; however, the required daily intraperitoneal injection and oral gavage are a significant burden to the neonatal mice. Other non-genetically induced mild SMA models include suboptimal dosing with AAV9-SMN (Meyer et al, 2015), oligonucleotides targeting SMN splicing (Zhou et al, 2015; Osman et al, 2016), and AAV-9s targeting disease-modifying proteins such as plastin-3 (Kaifer et al, 2017) and follistatin (Feng et al, 2016). Each intervention presents unique challenges for the studying of co-therapies. Stress resulting from repeated injections in neonatal mice may blunt synergies, and the CMV enhancer/chicken-*β*-actin promoter used to drive SMN in AAV-9–based interventions may not be consistently activated (Lukashchuk et al, 2016; Nieuwenhuis et al, 2020). Strong and constitutively activated promoters are prone to inactivation because of extensive methylation (Domenger & Grimm, 2019). SPINRAZA has an estimated terminal half-life of 135–177 d in the cerebrospinal fluid and 63–87 d in the plasma (Neil & Bisaccia, 2019), increasing the likelihood of drug–drug interactions. These challenges highlight the need for novel approaches to study co-therapies and to distinguish potential drug–drug interactions.

Our goal was to modify the severe Li SMA mice to a milder SMA mouse model with minimal intervention on the treated newborn that will allow efficacy testing of combinatorial therapies with limited drug–drug interactions. For these studies, we used the previously published human *SMN2* splicing modifier NVS-SM2, which promotes exon 7 inclusion and restores normal SMN protein expression, although less efficient in promoting exon 7 inclusion than Branaplam (NVS-SM1) at 3 mg/kg in C+/+ SMA mice (Palacino et al, 2015). The influence of NVS-SM2 on life span in SMA mice has not been reported. Pharmacokinetic analysis demonstrated that NVS-SM2 is readily available in the brain after IV and oral (PO) administration in mouse and rat with $T_{max}$ of 3 h after PO with 3 mg/kg in mice, and treatment induced a 1.5-fold increase in SMN protein levels in the mouse brain (Palacino et al, 2015). The advantage of a pharmaceutically induced mild SMA model in the Li SMA mice is their favorable breeding scheme with 50% of their progeny exhibiting symptomatology due to pathologically low levels of the SMN protein.

## Results

We synthesized NVS-SM2 and confirmed activity in our previously reported *SMN2* reporter cell assay (Fig S1A) (Cherry et al, 2012, 2013). NVS-SM2 caused a dose-dependent increase in SMN-luciferase expression up to 1,500%, followed by a decline at ~3 *μ*M due to cytotoxicity, as indicated by a decrease in Renilla luciferase (Fig S1A). To investigate in vivo activity, severe SMA mice were generated according to the breeding scheme that produces severe SMA mice (*Tg*(*SMN2*)*2Hungtg/0*; *Smn1tm1Hung/tm1Hung*) and heterozygous (Het, *Tg*(*SMN2*)*2Hungtg/0*; *Smn1tm1Hung/wt*) control siblings as described by Gogliotti et al (2010). Heterozygous (Het) mice express both mouse and human SMN protein, whereas severe SMA mice express only low levels of human SMN generated from the human *SMN2* transgene. Het and severe SMA neonatal mice were injected s.c. with 1 mg/kg NVS-SM2 or vehicle (PEG:PBS, 50:50) once daily for five consecutive days, beginning at PND 2. On PND 7, mice were euthanized and tissues harvested. SMN levels across treatment groups were quantified using the human SMN-specific monoclonal SMN antibody 2F1. Antibody specificity for human SMN was confirmed in whole brain lysates of Het, severe SMA, and non-transgenic FVB/NJ PND 7 old mice. For comparison, SMN protein levels were also investigated using MANSMA 6, which detects human and mouse SMN protein. The 2F1 antibody did not detect SMN in non-transgenic FVB/NJ mice, whereas MANSMA 6 detected SMN in all three strains (Fig S1B). NVS-SM2 treatment increased human SMN protein levels by 4.5-fold in brain ($P$ = 0.0005), and 2.5-fold in spinal cord ($P$ = 0.0355) and in muscle tissues in severe SMA mice (Figs 1A–C and S2A and B). Human SMN protein levels in the muscle of vehicle-treated severe SMA mice were not detectable (Fig 1C). In the Het control cohorts, we detected higher levels of human SMN in all tissues than severe SMA (Fig S2B). We hypothesize that the human SMN proteins expressed in the Het control mice are increased by mouse SMN proteins because of the oligomerization properties of SMN (Lorson et al, 1998) and/or through increased SMN exon 7 incorporation due to higher SMN protein levels (Jodelka et al, 2010; Ruggiu et al, 2012). NVS-SM2 treatment also increased SMN protein in brain, spinal cord and muscle in Het siblings. This extends the work by Palacino et al (2015) who reported a 1.5-fold increase in SMN protein in C+/+ mice brain tissues after a single 30 mg/kg NVS-SM2 administration (Palacino et al, 2015). These authors found that NVS-SM1 and NVS-SM2 have similar pharmacokinetic properties in mice and rats. However, NVS-SM1 resulted in a greater increase in *SMN2* splicing in C+/+ mice than NVS-SM2, whereas both increased SMN protein in brain tissue at 30 mg/kg oral treatment by 1.5-fold in C+/+ mice. In comparison, we show that a 30-times lower dose of NVS-SM2 administered daily s.c. for five consecutive days, increased brain SMN protein by 4.5-fold in severe SMA and Het mice. These differences in SMN expression may be due to the administration method or the mouse models used. In addition, NVS-SM2 induction of *SMN2* splicing may occur more slowly than with NVS-SM1.

As SMN protein levels increased in the central nervous system and the periphery, we evaluated the biological impact of NVS-SM2 on the temporal development and progression of the SMA phenotype. The severe SMA mice were injected s.c. with NVS-SM2 at 0.1

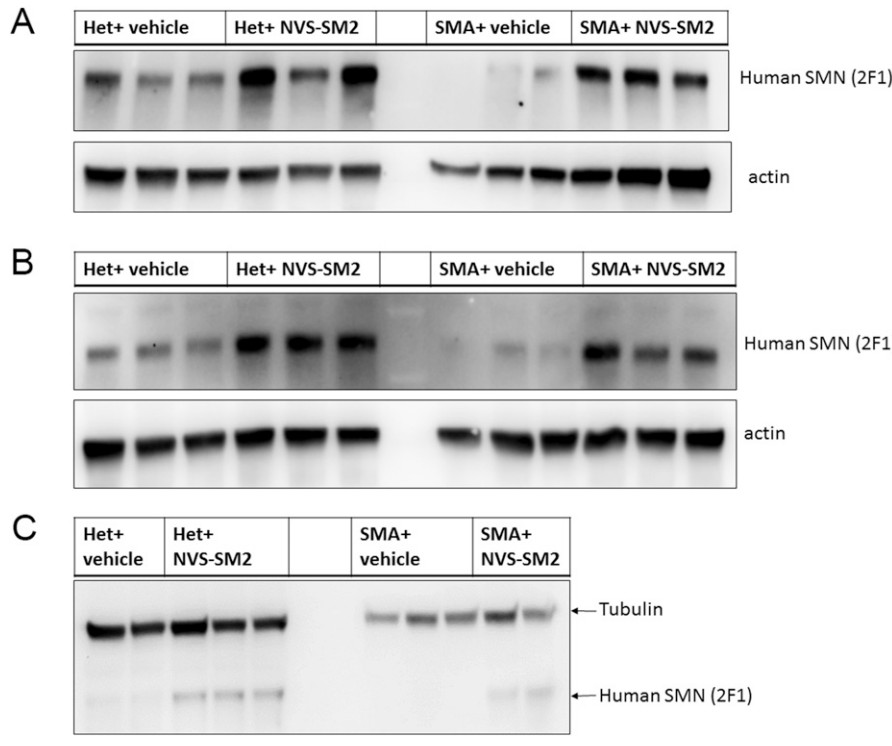

**Figure 1.  NVS-SM2 increases survival motor neuron (SMN) protein in severe 5058 spinal muscular atrophy (SMA) and control Het mice.**
Mice were injected s.c. with 1 mg/kg NVS-SM2 or vehicle starting PND 2 daily until PND 6. Mice were euthanized and tissues harvested on PND 7. **(A, B, C)** Human SMN protein levels were analyzed in brain (A), spinal cord (B), and muscle (C) tissues via immunoblotting. SMN protein was normalized to actin and tubulin. Each lane represents tissue from an individual mouse. SMA represents the severe 5058 SMA mice, which express the human *SMN2* transgene, and Het mice are their littermates, expressing both human and mouse SMN.

and 1 mg/kg daily from PND 2–15, followed by every other day until PND 30. All mice were alive at PND 30, demonstrating a successful intervention as severe SMA mice typically succumb by PND 11. Hence, we stopped compound delivery and continued monitoring the mice for lifespan and any phenotypic changes. We observed a median survival of PND 57 and PND 94 after a 30-day treatment with 0.1 and 1 mg/kg, respectively (Fig 2A). The surviving mice were mobile, but progressive distal limb, tail, and ear necrosis necessitated euthanasia (Video 1). Kaplan–Meier survival curves with the Mantel–Cox log rank test demonstrated that both treatment dosages significantly differed from the vehicle treatment (VH versus 0.1 mg/kg: *P* = 0.0036; VH versus 1 mg/kg: *P* = 0.0014). Heterozygous control mice and NVS-SM2–treated severe SMA mice weighed ~20 and 15 g, respectively, by PND 30, and weight began to decline at PND 63 in the 1 mg/kg–treated severe SMA group (Fig 2B). Ear necrosis emerged at PND 73. To statistically compare the phenotype between groups, we chose the time point at which the treatment group gained its maximum average weight (MAW). For the 1 mg/kg treatment group, the MAW was at PND 63 with no significant difference in weight between severe SMA mice and Het controls. As a phenotypic marker, we also measured tail length, which was slightly shorter in untreated SMA mice than in Het mice, and correlated with body weight (Figs 2C and S3A). Tail length in the 30-days 1 mg/kg group was slightly shorter than in the Het group, but the difference was not significant (PND 63: 1 mg/kg versus Het versus: 7.33 ± 0.17 versus 8.13 ± 0.24 cm; *P* = 0.054, Fig 2D). The MAW for the 0.1 mg/kg group was reached at PND 32, and low-dose–treated mice were significantly lighter than age-matched Het controls (PND 32: 0.1 mg/kg versus Het: 15.23 ± 2.47 g versus 20.1 ± 0.79; *P* = 0.0296).

This optimized low-dose 30-d treatment regimen may represent a tractable mild SMA mouse model that could resemble the phenotypic delay in human Type II/III SMA patients. Both doses of s.c. NVS-SM2 greatly improved phenotypic outcomes compared with oral administration of NVS-SM1 in SMNΔ7 mice (Palacino et al, 2015). Daily oral administration of NVS-SM1 at 3 mg/kg was reported to rescue 60% of SMA mice at PND 30. We predict that treatment with 1 mg/kg NVS-SM2 past PND 30 will yield the same body weight and tail length in severe SMA as in Het mice. In addition, continuous treatment at 0.1 mg/kg may suffice for full rescue.

Based on this unexpected long-term rescue with very low doses of NVS-SM2, we investigated whether a shorter treatment duration would result in a moderate SMA model in severe SMA mice. Neonatal mice were injected s.c. once daily with 1 mg/kg NVS-SM2 on three consecutive days (PND 2–4). Surprisingly, the median survival increased to 30 d, and body weight averaged 10 g, ~70% of age-matched Het control weight (Fig 3A and B). The MAW at PND 25 was significantly different between Het versus 1 mg/kg (Het versus 1 mg/kg: 10.5 ± 0.67 versus 15.4 ± 0.47 g; *P* = 0.0002). We then decreased the concentration of NVS-SM2 to 0.1 and 0.5 mg/kg and repeated the treatment regimen. These mice showed a dose-dependent decline in median survival to PND 18.5, which is 2.5 d before weaning, and PND 26, respectively. The survival curves of all treatment groups were significantly different from untreated control severe SMA mice (VH versus 0.1 mg/kg: *P* = 0.0004; VH versus 0.5 mg/kg: *P* = 0.0004; VH versus 1 mg/kg: *P* = 0.0004, Fig 3A). Mice injected with the lowest dose exhibited the lowest gain in body weight and reached their MAW at PND 14 (PND 14: Het versus 0.1 mg/kg: 9.9 ± 0.3 versus 6.7 ± 0.4 g; *P* < 0.0001; Fig 3B), whereas the 0.5 mg/kg–treated mice reached their MAW at PND 22 (PND 22: Het versus

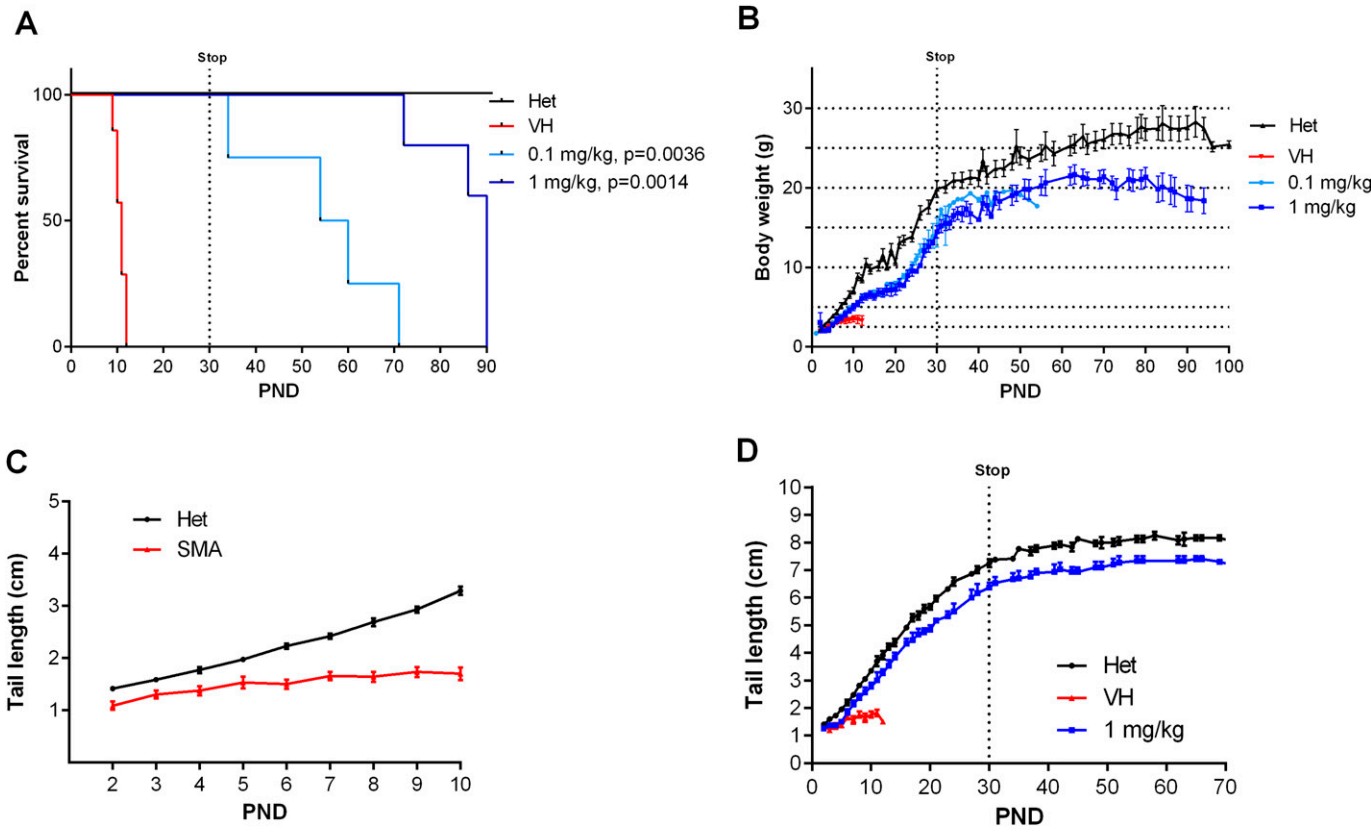

**Figure 2.  Effect of different doses of NVS-SM2 on severe 5058 spinal muscular atrophy (SMA) mice.**
**(A)** Kaplan–Meier survival curves of severe 5058 SMA mice s.c. treated with vehicle (n = 7) or 0.1 (n = 4) and 1 mg/kg (n = 5) NVS-SM2 daily until PND 15 and then every other day until PND 30 (A). Mantel–Cox test was used to analyze survival differences between NVS-treated and SMA mice and *P*-values are presented in the legend. **(B, C, D)** Mice were monitored for body weights (B) and tail length (C, D). Data expressed as SEM.

0.5 m/kg: 12.3 ± 0.3 versus 8.9 ± 0.3; *P* < 0.0001; Fig 3B). To test short-term treatment with an even further reduced drug amount, we injected 0.1 mg/kg s.c. for only 2 d: PND 2 and PND 3. This introduced greater variability, with a median survival of 13 d and one mouse surviving until PND 24, rendering this treatment schedule unsuitable (Fig 4A). Body weights were only marginally improved (Fig 4B).

To investigate whether NVS-SM2 is effective when administered orally, we treated mice with 1 mg/kg for 3 d (PND 2–4). These mice had a median life span of 29 d (VH versus 1 mg/kg p.o.: *P* = 0.0024, Fig 3A). The MAW was reached at PND 18 with an average weight of 7.5 ± 1.2 g. This included a runt with a weight of 1.1 g on PND 2 that grew to 4 g at PND 18, whereas the remaining treated severe 5058 SMA mice had an average weight of 1.9 g on PND 2 and reached 8.7 ± 0.4 g on PND 18 (Fig 3B). Despite the low birth weight and the small volume (2.2 μl) of NVS-SM2 administered, the successful response to the oral drug delivery was remarkable. Tail length in the 0.5 mg/kg group peaked at PND 16 at 3 cm, and tails were much thinner. Tail necrosis followed tail thickening and was overt at PND 23. Tail length of mice treated with the lowest dose peaked at PND 14 at 2.4 cm. These animals were found dead before appearance of tail necrosis (Fig 3C and D), although their tails were notably thinner. This coincides with the observation that the tails of 5058 SMA mice with two copies of *SMN2* become necrotic after weaning (PND 21). Although the ratio of tail length to body weight was not significantly

different in severe SMA versus Het mice (Fig S3A), there was a significant difference in the 3-d treatment groups (0.5 mg/kg s.c. and 1 mg/kg PO) compared with Het mice (Fig 3D), revealing tail length as a useful and early phenotypic marker of rescue. With disease progression, these severe mouse models exhibit necrosis of the ears, tail, and digits because of vascular thrombosis. Digital necrosis has been also reported in a small number of infants with severe SMA (Araujo et al, 2009; Rudnik-Schoneborn et al, 2010). Mice were analyzed for gross motor function using the beam/pen test starting at PND 12. On average, treated severe 5058 SMA mice performed as well as Het control mice (Fig 3E).

Because SMA research is commonly conducted using the SMNΔ7 transgenic mouse strain, we also investigated the outcome of the 3-d treatment regimen in these mice. NVS-SM2 was injected once daily s.c. at 1 mg/kg on PND 2–4, resulting in a median survival of 29.5 d compared with 14.5 d in vehicle-treated mice (*P* = 0.004, Fig 5A). Body weights were improved and peaked at PND 16 (Fig 5B). These results are comparable to those observed in NVS-SM2–treated severe SMA (5058) mice, demonstrating this approach to be suitable across SMA mouse models. The impressive potency of NVS-SM2 appears to surpass NVS-SM1, which rescued ~60% of SMNΔ7 mice after daily oral administration at 3 mg/kg (Palacino et al, 2015).

Because of the in vivo potency of NVS-SM2, we investigated next the suitability of NVS-SM2 to develop an animal model allowing for

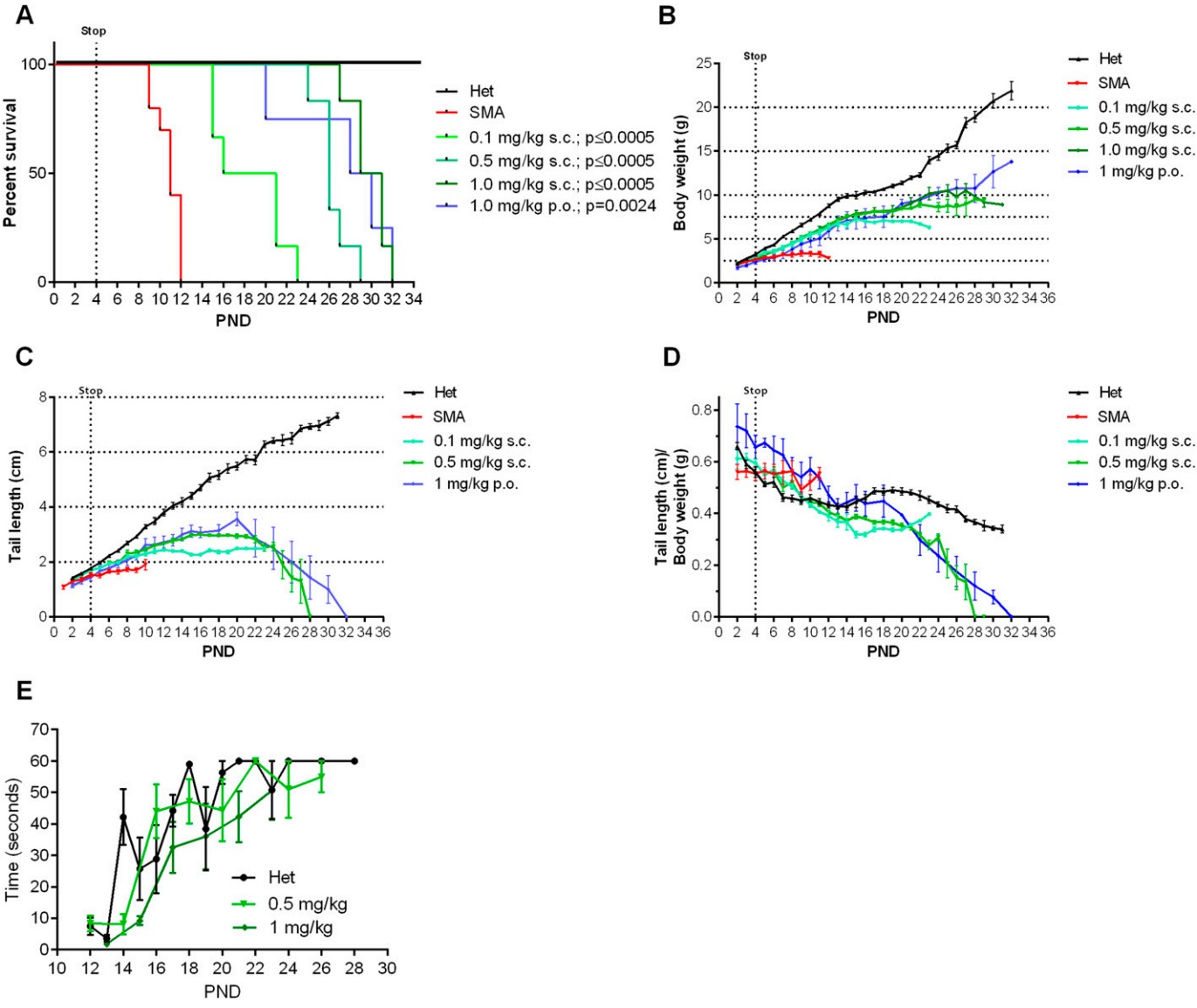

**Figure 3. Effect of 3-d treatments with different dosages of NVS-SM2 on severe 5058 spinal muscular atrophy (SMA) mice.**
Severe 5058 SMA mice were treated on PND 2, PND 3 and PND 4 s.c. or orally (p.o.) with the indicated doses of NVS-SM2. **(A)** Kaplan–Meier survival curve of severe 5058 SMA mice s.c. treated with 0.1 (n = 6), 0.5 (n = 6), and 1 mg/kg (n = 6) NVS-SM2 or orally with 1 mg/kg NVS-SM2 (n = 4) and untreated SMA mice (n = 10). Mantel–Cox test was used to analyze survival differences between NVS-treated and SMA mice and *P*-values are presented in the legend. **(B, C, D)** Mice were monitored for body weights (B) and tail length (C) to determine the tail length: body weight ratio (D). **(E)** Pen test time of 0.5 mg/kg NVS-SM2–treated mice in comparison to Het mice (E). Data expressed as SEM.

the initiation of therapeutic intervention during the symptomatic stage. We performed a trial experiment with NVS-SM2 s.c. at 1 mg/kg starting at PND 6, a time point considered symptomatic in 5058 SMA mice (Groen et al, 2018). Mice received daily drug injections until PND 15, then every other day until PND 30, at which point treatment was stopped. Mice had a median survival of 70 d, slightly lower than the median survival of 94 d achieved with PND 2 treatment (*P* = 0.0063, Fig 6A). Therefore, we investigated if long term survival could be achieved at late symptomatic stages of the disease. We chose PND 8 as the last feasible time point because severe 5058 SMA mice have a median life span of 11 d, with some living only until PND 9, rendering PND 9 as an impractical starting point. Groen et al (2018)

reported that at PND 8, SMN protein is decreased systemically by two to threefold compared with symptomatic (PND 5) and control litter mates (Groen et al, 2018). Mice were injected with 1 mg/kg, s.c. once per day beginning at PND 8. This protocol generated two distinct outcomes: short-term and long-term survivors. The mice that did not respond succumbed with a median life span of 12 d; survivors continued to thrive (Video 2), and injections were continued four times per week. The experiment was stopped at PND 110 (Fig 6B). The body weight of the surviving NVS-SM2–treated mice averaged ~80% of age-matched Het controls (Fig 6C). The tail was significantly shorter in all survivors (Figs 6D and S3B). Body weight at treatment start (PND 8) did not influence the outcome (Fig 7B).

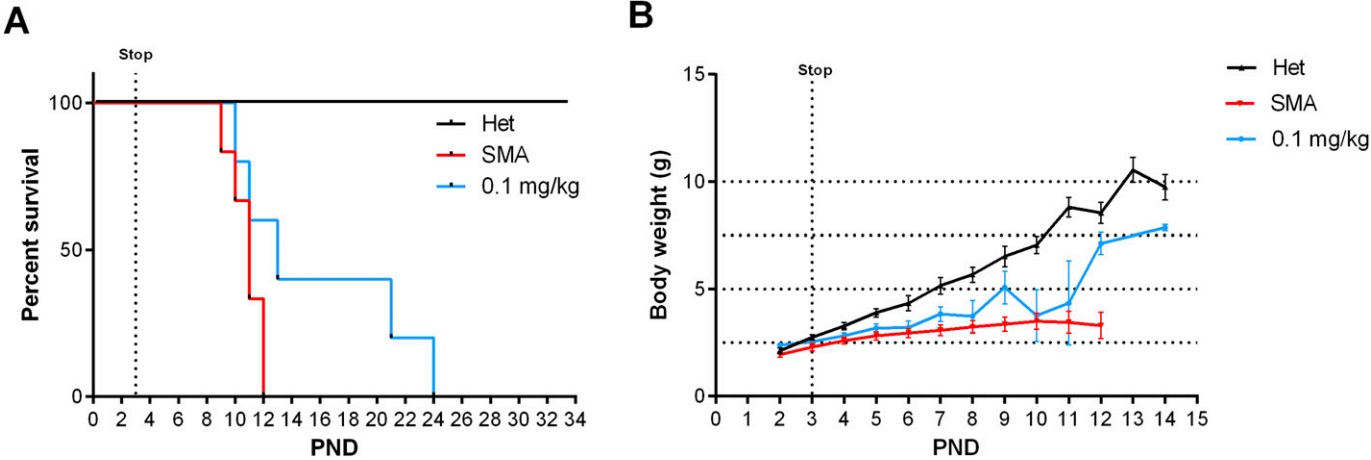

**Figure 4. Impact of NVS-SM2 treatment on severe 5058 spinal muscular atrophy (SMA) mice survival after 2-d treatments.**
Severe SMA mice were treated on PND 2 and PND 3 with the indicated doses of NVS-SM2. **(A)** Kaplan–Meier survival curve of severe 5058 SMA mice treated s.c. with 0.1 mg/kg NVS-SM2 (n = 5) on PND 2 and PND 3 in comparison to untreated SMA mice (n = 6). Mantel–Cox test was used to analyze survival differences between NVS-treated and SMA mice and *P*-values are presented in the legend. **(B)** Mice were monitored for body weights. Data expressed as SEM.

Small spleen size is a hallmark of the SMA phenotype in mice but not in humans (Khairallah et al, 2017). Spleen weights were lighter than control spleens, although the difference was not significant (SMA versus Het: 89.2 ± 8.1 versus 104 ± 3.8 mg) and no difference was observed when spleen weight was normalized to body weight (Fig 7A). SMN protein levels in the brain and spinal cord in long-term survivors were similar to that of control Het mice (Fig 8A–D). Our study, together with Risdiplam data, indicates the potential benefit of splicing-modifier drugs later in disease in both severe and mild SMA. In addition, we show that there is a treatment window beyond PND 7 in the severe 5058 SMA mice. Additional regimens with initial 3-d drug exposure and later stage pulses can be considered to evaluate disease progression in SMA mice and the rescue capabilities of NVS-SM2, using weight gain, tail length, and the emergence of digital necrosis as indicators of disease progression.

## Discussion

Here, we report that the *SMN2* splicing modifier NVS-SM2 is highly active in vivo and could be titrated in dosage, timing, and duration of administration for the development of robust SMA mouse models with varying stages of disease. NVS-SM2 increased SMN protein in the brain, spinal cord, and muscle tissues in severe SMA (5058) mice and extended their lifespan and bodyweight. s.c. and oral administration for 3 d (PND 2, 3, and 4) was sufficient to significantly extend survival in a dose-dependent manner. The limited invasiveness of this treatment model eases the stress on the neonatal mice, providing a significant advantage over daily injections, and potential drug–drug interactions are eliminated because of the brief duration of drug exposure. NVS-SM2 rescued also severe SMA mice at late symptomatic time points. Continuous treatment

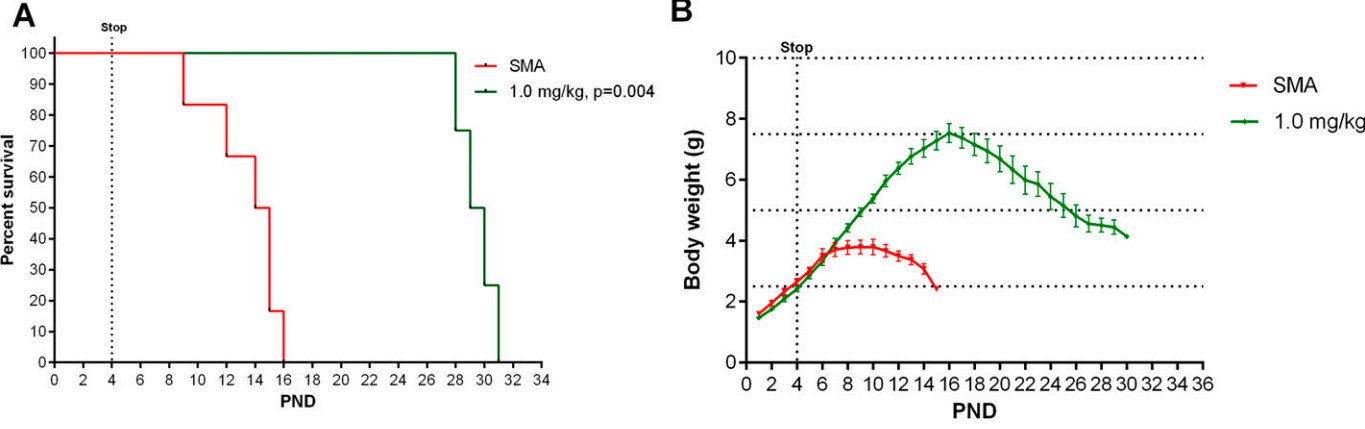

**Figure 5. Effect of 3-d treatments with NVS-SM2 on SMNΔ7 mice.**
SMNΔ7 mice were treated on PND 2, PND 3, and PND 4 s.c. with 1 mg/kg NVS-SM2. **(A, B)** Kaplan–Meier survival curve (A) and body weights (B) of untreated (n = 10) and NVS-SM2 (n = 4)–treated SMNΔ7 mice. Mantel–Cox test was used to analyze survival differences between NVS-treated and untreated SMA mice and *P*-values are presented in the legend. Data expressed as SEM.

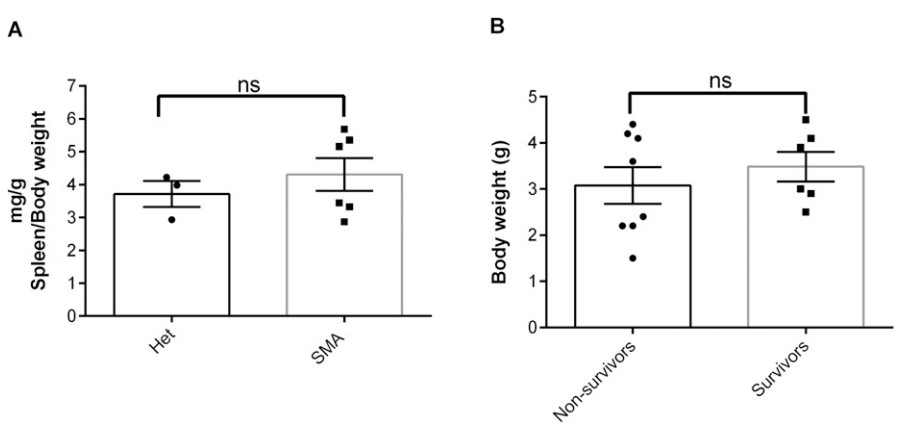

**Figure 6. Effect of late treatment of severe spinal muscular atrophy (SMA) (5058) mice with NVS-SM2.**
Animals were treated starting on PND 6 (A, n = 3) or PND 8 (B, n = 14) s.c. with 1 mg/kg NVS-SM2. **(A, B)** Kaplan–Meier survival curves. PND 8–treated SMA mice were further separated into non-survivors (n = 8) and survivors (n = 6) groups. Mantel–Cox test was used to analyze survival differences between NVS-treated and SMA mice and *P*-values are presented in the legend. **(C, D)** PND 8–treated severe 5058 SMA mice were monitored for body weights (C) and tail lengths (D). Data expressed as SEM.

starting at PND 8 resulted in a group that survived until the end of the experiment at PND 110. This treatment starting point has not been therapeutically beneficial with antisense oligonucleotide or adeno-associated virus-9 SMN gene therapy. We speculate that both systemic exposure and the dramatic increase in SMN protein

levels contribute to this outcome. Indeed, SMN protein levels in the brain and spinal cord were comparable to Het control mice (Fig 8A–D). Future studies are required to assess the status of other phenotypic SMA markers such as NMJ and muscle pathology in these late-stage rescued animals and the mechanisms underlying

**Figure 7. Effect of late treatment of NVS-SM2 in severe 5058 spinal muscular atrophy (SMA) mice.**
**(A)** Spleen weights of NVS-SM2–treated severe 5058 SMA mice at PND 110 in comparison to healthy Het control mice are not different (A). **(B)** Average body weight at PND 8 of the treated NVS-SM2 mice separated into the survivors and non-survivors groups are not different (B). Data expressed as SEM and analyzed using Student's unpaired *t* test, a *P*-value of 0.05 was taken as significant. ns indicates nonsignificance.

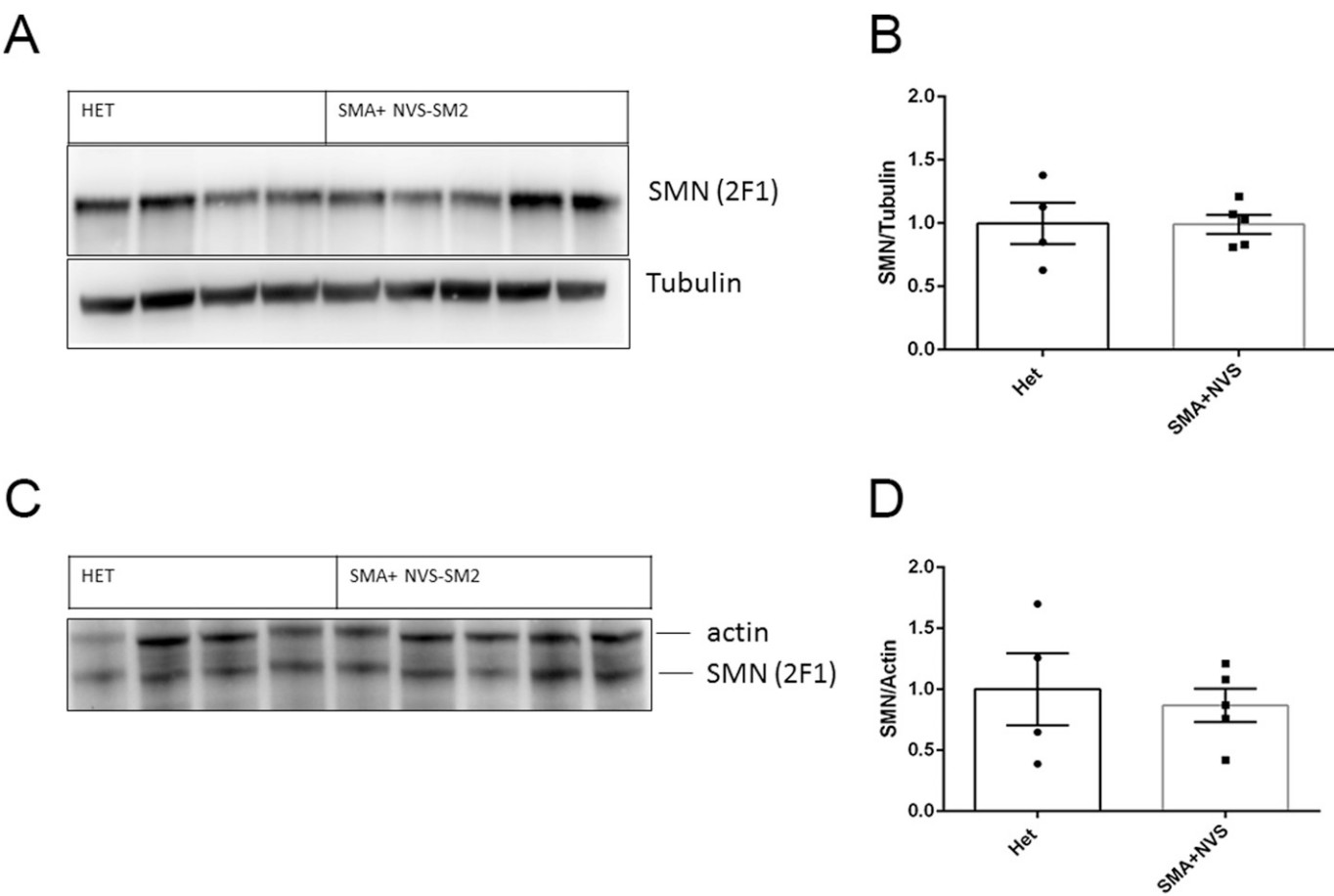

**Figure 8.  Survival motor neuron (SMN) protein levels in NVS-SM2–treated severe 5058 spinal muscular atrophy (SMA) mice at PND 110 compared to healthy Het control mice.**
**(A, B, C, D)** Immunoblot of human SMN and housekeeping proteins at PND 110 in Het (n = 4) and NVS-SMN2 PND 8–treated severe SMA 5058 mice (n = 5) in brain (A, B) and spinal cord (C, D). SMN protein was normalized to housekeeping proteins (B, D). Each lane represents tissue from an individual mouse. Data expressed as SEM and analyzed using Student's unpaired $t$ test, a $P$-value of 0.05 was taken as significant.

their recovery. Of particular importance will be to investigate how long the SMN induction by NVS-SM2 lasts because the drug has a half-life of 3 h, yet we observed a lifespan extension of 26 d past the last administration. This effect provides a significant opportunity to investigate the temporal requirements of SMN in the pathogenesis of SMA.

Most post-symptomatic rescue experiments in severe SMA mice have been performed with antisense oligonucleotides (ASO) and disease-modifying gene therapies (Foust et al, 2010; Le et al, 2011; Gogliotti et al, 2012; Robbins et al, 2014). The critical therapeutic window in SMNΔ7 mice is PND 5–6 (Foust et al, 2010; Le et al, 2011; Gogliotti et al, 2012; Robbins et al, 2014). SMNΔ7 mice, on average, survive 4 d longer than severe 5058 SMA mice. SMNΔ7 mice injected intracerebroventricularly (ICV) with scAAV9-SMN on PND 8 had a median survival of 18 d compared with 14 d for the non-injected mice. Mice injected a day earlier showed a greater response with a median survival of 28 d, and a few mice survived for 70 d (Robbins et al, 2014). IV injection of scAAV9-SMN on PND 5 modestly increased survival (30 d). Injection at PND 10 did not rescue early lethality. Inducible transgenic rescue experiments also point to PND 6 as the

latest time point to rescue SMNΔ7 mice (Le et al, 2011; Lutz et al, 2011). However, a study in a pharmacologically induced mild SMNΔ7 mice model reported that mice treated with suboptimal dosing of SMN-C3 changed to optimal dosing at a later stage (PND 32) had increased body weight and improved phenotypic markers compared with mice that continued to receive the suboptimal dose (Feng et al, 2016). Few post-symptomatic rescue experiments have been conducted in severe 5058 SMA mice. The study by Hua et al (2011) used a late treatment protocol with ASO treatments on PND 5 and a second on PND 7, which resulted only in a modest increase of survival to PND 16 and a few animals lived past PND 100 (Hua et al, 2011). The maturation of the neonatal blood–brain barrier may restrict ASO access. These studies differ from our findings that NVS-SM2 treatment rescues SMA mice beyond PND 40 and treatment on PND 8 results in a group of survivors that thrive albeit with a shorter tail.

Although we have not observed any defects in gross motor function with the 3-d dosing regimen, it is possible that with a more sensitive test, for example, a rotarod performance test, a defect

becomes apparent in this intermediate 30-d survival model. The 5058 severe SMA mice are reported to selectively lose α-motor neurons but not γ-motor neurons, which coincides with reduced α-motor neuron cell body area in all spinal cord regions (Powis & Gillingwater, 2016). In addition, these SMA mice develop denervation of intercostal, transversus abdominis and appendicular FDB-2/3 muscles similar to the SMNΔ7 SMA mouse model (Murray et al, 2008; Ling et al, 2012; Keil et al, 2014; Lin et al, 2016). Although that we have not assessed motor neuron integrity post-treatment in these severe SMA models, both have motor neuron defects of varying degrees with denervation being more pronounced in the SMNΔ7 SMA mice than in the Taiwanese mice at disease end-stage (Ling et al, 2012; Lin et al, 2016; Powis & Gillingwater, 2016). However, in SMNΔ7 SMA mice, SMN overexpression limited to motor neurons (McGovern et al, 2015) or the majority of most of the neurons using a synapsin promoter (Besse et al, 2020), only partially rescued survival, inferring a degree of dependency on non-neuronal SMN expression. The requirement for SMN outside the nervous system was implied by sustained responses with s.c. administration of antisense oligonucleotide and induction of SMN in the Taiwanese SMA model (Hua et al, 2011). Nonetheless, in humans the non-neural consequences of low SMN protein is less established. The accepted disease driver in human SMA is motor neuron death, although there are anecdotal reports that peripheral organs such as heart, vasculature, muscle, pancreas and liver may be also impacted in SMA patients (Lipnick et al, 2019; Hensel et al, 2020). We anticipate that future studies can now be performed throughout the disease course in these pharmacologically manipulatable SMA mice to determine the functions of SMN that maintain motor neuron integrity and muscle innervation.

# Materials and Methods

## Materials and antibodies

PEG400 and DMSO were purchased from Sigma-Aldrich. PBS, DMEM, Pen/strep was purchased from Gibco. Fetal bovine serum was purchased from Peak serum. The following antibodies were used: anti-SMN (1:2,000; 2F1; Cell Signaling Technologies and MANSMA 6, 4H2; Developmental Studies Hybridoma Bank, DSHB), anti-tubulin (DM1α, 1:4,000; Sigma-Aldrich), and anti-β actin (AC-74, 1:4,000; Sigma-Aldrich). MANSMA6 (4H2) was deposited to the DSHB by Morris, GE (DSHB Hybridoma Product MANSMA6 (4H2)).

## NVS-SM2 preparation

NVS-SM2 (2-(6-(methyl(2,2,6,6-tetramethylpiperidin-4-yl)amino) pyridazin-3-yl)-5-(1H-pyrazol-4-yl)phenol) was prepared following the procedures described in patent (Cheung et al, 2012). Liquid chromatography–mass spectrometry and $^1$H NMR of the final NVS-SM2 were consistent with its structure and the published data (Palacino et al, 2015). NVS-SM2 was dissolved in 100% DMSO (vol/vol) for the cell culture experiments. For the in vivo experiments, NVS-SM2 was dissolved in PEG400 by vortexing followed by addition of PBS resulting in 50:50 percent PEG400:PBS.

## Cell culture and reporter cell assay

SMN2 reporter cells were grown in DMEM containing 10% FBS and 1× Pen/Strep. The assay was completed as previously described (Rietz et al, 2017). In brief, cells were seeded in a 96-well plate at 25,000 cells/well. The following day cells were treated with threefold serial dilutions of compounds, incubated for 24 h, lysed, and analyzed using the Dual-Glo Luciferase Assay System (Promega).

## Mice breeding, genotyping, and treatments

The animal study protocols were approved by the Institutional Animal Care and Use Committee of Indiana University and conform to the Guide for the Care and Use of Laboratory Animals. Study protocols were also approved by the University of Missouri Animal Care and Use Committee as well as the regulations established by the National Institute of Health's Guide for the Care and Use of Laboratory Animals. Severe SMA (5058) neonatal mice were bred as previously reported (Gogliotti et al, 2010). Animals were maintained on a 12–12 h light–dark cycle with food and water ad libitum and were provided with Bed-r'Nest as standard of care. DNA was isolated from tail snips of ~0.1–0.15 cm length using the QIAGEN DNAeasy Kit. $SMN1^{tm1}$ and SMN2 genotyping was performed as directed by Jackson Laboratory using their suggested primers. Treatment started on postnatal day 2 (PND 2) unless otherwise stated, with PND 0 as the day mice were born. All treatments used the same vehicle (PEG400: PBS (50:50)) to solubilize NVS-SM2. Because of the high drug potency, 1 mg/ml was prepared and serially diluted to obtain final stock solutions of 0.1–0.01 mg/ml. Mice were treated with a final dose of either 0.1, 0.5, or 1 mg/kg via daily s.c. injection or oral administration. s.c. administration was performed at 10 μl/g body weight, and oral administration at 2 μl/g body weight. Oral administration in neonates was performed using a plastic feeding tube (FTP-20-30-50; Instech Laboratories). Feeding tubes were placed in the inner-cheek, and the suspension was applied slowly, stimulating suckling behavior. In brief, the following treatment schedules were used (i) 30-d treatments started on PND 2, everyday s.c. until PND 15 followed by every other day until PND 30, (ii) 3-d treatment groups received treatment on PND 2, PND 3, and PND 4 with the indicated doses (iii) 2-d treatment groups on PND 2 and PND 3 (iv) PND 6 late treatment: starting PND 6, daily until PND 15 from then followed by every other day until PND 30, and (v) PND 8 late treatment: starting PND 8, daily until PND 15 from then four times a week (2 d on, 1 d off, 2 d on, 2 d off, and repeat) until PND 110. Mice were euthanized via $CO_2$ exposure and the whole brain, spinal cord, spleen, and muscle (left vastus lateralis) were extracted for Western blotting. We used the balance beam/pen test to assess motor balance and coordination in treated SMA mice, as described in SMA_M.2.1.001 published by TREAT-NMD Neuromuscular Network. The beam/pen test was conducted every other day from PND 12 until PND 30. For the severe SMNΔ7 animal model, heterozygous breeder pairs of mice ($Smn^{+/−};SMN2^{+/+};SmnΔ7^{+/+}$), were purchased from the Jackson Laboratory (JAXStock#005025:FVB.CgGrm7Tg (SMN2)89AhmbSmn1tm1MsdTg (SMN2*delta7)4299Ahmb/J). The colony was maintained as heterozygote breeding pairs under specific pathogen free conditions. Experimental mice litters ($Smn^{−/−};SMN2^{+/+};SMNΔ7^{+/+}$ referred as SMNΔ7) were genotyped on the day

of birth (PND 0) using standard PCR protocol (JAX Mice Resources) on tail tissue material as previously described (Osman et al, 2019). Experimental pups were kept with a minimum of two healthy heterozygous siblings.

## Tissue isolation and immunoblotting

Extracted whole brain, spinal cord, and vastus lateralis muscle tissues were extracted on PND 7 and lysed in the pre-heated lysis buffer (2% SDS, 150 mM NaCl, 10 mM Tris–HCl, pH 8.0, + 1× Pierce protease inhibitor cocktail [added fresh, A32961]) at a tissue:buffer volume ratio of 1:15. Tissues were heated in lysis buffer for 5 min at 95°C. Brain and spinal cord were disrupted and homogenized using a 22-gauge needle, whereas muscle tissue was first homogenized using an 18-gauge needle followed by a 22-gauge needle. Lysates were then heated for 10 min at 95°C, and cleared by centrifugation at 8,000$g$ for 10 min. Protein concentration was measured using the Pierce BCA kit, and equal amounts of protein were separated on a 4–12% SDS gel (Genscript SurePAGE). Proteins were transferred onto a polyvinylidene (PVDF) Immobilon-P (0.45 $\mu$m; Millipore), and SMN, $\beta$-actin, and $\alpha$-Tubulin proteins were visualized by chem-iluminescence after exposure to the anti-SMN antibody (2F1; Cell Signaling) following by an HRP-linked secondary antibody. Signal intensities were quantified using ImageJ and Image Studio Lite (LI-COR Biosciences). SMN expression is expressed as fold change and normalized to housekeeping protein, $\beta$-actin, and $\alpha$-Tubulin as indicated.

## Data analysis and statistics

Survival was analyzed with Kaplan–Meier survival curves using the log-rank Mantel–Cox test for survival comparisons (Graph-Pad Prism v6.00; GraphPad Software, Inc.). A $P$-value of $P < 0.05$ was considered statistically significant. The MAW is defined as the MAW at the last time point at which all mice of the treatment group that entered the study were still alive. This time point was used to determine if body weights or tail length differences were statistically different. Groups of two were analyzed using Student's unpaired $t$ test, and groups of more than two were analyzed using one-way ANOVA with post-hoc analysis (Dunnett's or Bonferroni as indicated). All data are expressed as SEM unless otherwise stated.

# Supplementary Information

# Acknowledgements

This research was supported by grants National Institute of Neurological Disorders and Stroke (NINDS) R33NS095139 and CureSMA to EJ Androphy, CL Lorson, and KJ Hodgetts. The content is solely the responsibility of the authors. We thank Jacob Astroski for assistance with mouse experiments, and Academia Sinica for providing the 5058 mice. We also thank Developmental Studies Hybridoma Bank (DSHB) for the MANSMA6 (4H2) antibody, developed by Dr. GE Morris and obtained by DSHB, created by the Eunice Kennedy Shriver National Institute of Child Health and Human Development of the National Institutes of Health and maintained at The University of Iowa, Department of Biology, Iowa City, IA.

## Author Contributions

A Rietz: conceptualization, data curation, formal analysis, validation, investigation, visualization, methodology, and writing—original draft, review, and editing.
KJ Hodgetts: conceptualization, resources, formal analysis, supervision, funding acquisition, investigation, methodology, and writing—review and editing.
H Lusic: data curation, formal analysis, investigation, methodology, and writing—review and editing.
KM Quist: data curation and writing—review and editing.
EY Osman: data curation, formal analysis, investigation, methodology, and writing—review and editing.
CL Lorson: resources, formal analysis, supervision, funding acquisition, investigation, and writing—review and editing.
EJ Androphy: conceptualization, resources, formal analysis, supervision, funding acquisition, validation, investigation, project administration, and writing—original draft, review, and editing.

## Conflict of Interest Statement

CL Lorson is the co-founder and CSO of Shift Pharmaceuticals.

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
