## [Reviewer comments · Life Science Alliance]

Life Science Alliance

Short duration splice promoting compound enables a tunable mouse model of spinal muscular atrophy

Anne Rietz, Kevin Hodgetts, Hrvoje Lusic, Kevin Quist, Erkan Osman, Christian Lorson, and Elliot Androphy

DOI: <https://doi.org/10.26508/lsa.202000889>

Corresponding author(s): Elliot Androphy, Indiana University School of Medicine

Review Timeline:

Submission Date:	2020-08-21
Editorial Decision:	2020-10-01
Revision Received:	2020-10-30
Editorial Decision:	2020-11-02
Revision Received:	2020-11-06
Accepted:	2020-11-09

Scientific Editor: Shachi Bhatt

Transaction Report:

October 1, 2020

Re: Life Science Alliance manuscript #LSA-2020-00889-T

Dr. Elliot J Androphy
Indiana University School of Medicine
Department of Dermatology 545 Barnhill Dr. Emerson Hall 139
Emerson 138
Indianapolis, IN 46202

Dear Dr. Androphy,

Thank you for submitting your manuscript entitled "Short duration splice promoting compound enables a tunable mouse model of spinal muscular atrophy" to Life Science Alliance (LSA). We apologize for this extended delay in getting back to you.

The manuscript has been reviewed by the editors and outside referees (reviewer comments below). As you will see, the reviewers were enthusiastic about the study and its potential impact, but have raised some concerns that should be addressed prior to further consideration of the manuscript at LSA. Therefore, although we are unable to publish the current version of the manuscript, we encourage you to submit a revised version that addresses these concerns.

Most of the concerns raised by the reviewers should be straightforward to address. A consensus has emerged, both from reviewers' comments and an internal anonymous conversation between the reviewers that it is crucial to clearly state what this model represents compared to other currently used models, and what aspects of human phenotype does this mouse model vs. what it doesn't. As for the request from Rev 1 about whether these intermediate mice do have some form of motor neuron involvement - we encourage you to provide some staining analysis that answers these concerns - if the stainings are not easily available we encourage you to address this point in discussion. Similarly, we also encourage you to elaborate on the fact that late introduction of therapy at 8 days gives a different result than Delta 7 mice.

We would be happy to discuss the individual revision points further with you, should this be helpful. A revised manuscript may be re-reviewed, most likely by some or all of the original referees.

When submitting the revision, please include a letter addressing the reviewers' comments point-by-point and a copy of the text with alterations highlighted (boldfaced or underlined). The typical time frame for revisions is three months. In an effort to expedite the review process, papers are generally considered through only one revision cycle.

Please use the link below to log into your account and submit your revised manuscript
<https://lsa.msubmit.net/cgi-bin/main.plex>

Thank you for considering Life Science Alliance (LSA) as an appropriate venue for your research. Please reach out to me if you have any questions.

Sincerely,

Shachi Bhatt, Ph.D.
Executive Editor
Life Science Alliance
<https://www.life-science-alliance.org/>
Tweet @SciBhatt @LSAJournal

- A letter addressing the reviewers' comments point by point.
- An editable version of the final text (.DOC or .DOCX) is needed for copyediting (no PDFs).
- High-resolution figure, supplementary figure and video files uploaded as individual files: See our detailed guidelines for preparing your production-ready images, <https://www.life-science-alliance.org/authors>
- Summary blurb (enter in submission system): A short text summarizing in a single sentence the study (max. 200 characters including spaces). This text is used in conjunction with the titles of papers, hence should be informative and complementary to the title and running title. It should describe the context and significance of the findings for a general readership; it should be written in the present tense and refer to the work in the third person. Author names should not be mentioned.

B. MANUSCRIPT ORGANIZATION AND FORMATTING:

Reviewer #1 (Comments to the Authors (Required)):

This manuscript by Rietz et al. explores both the utility of generating SMA mice of varying disease for future co-therapy studies and extend in vivo efficacy of a SMN2 splice modulator. The manuscript is well written, appropriately detailed and fills a gap the development of drug-induced milder SMA mice that have prolonged survival that are not extremely cumbersome to develop. The

comments provided are very minor and provided to enhance details for the reader and replication, extension studies that further explore this model and compound for developing milder SMA mice.

Introduction

1. Please update the following sentence to note the recent FDA approval of RisdiplamTM. "SMN2 splicing modifiers Risdiplam{trade mark, serif} and Branaplam{trade mark, serif} are in Phase 3 clinical trials for SMA type I (NCT02913482) and II (NCT02913482) and Phase 2 for type I (NCT02268552), respectively."
2. While describing the review by Dangouloff & Servais (2019) of clinical trial results it notes "...from the drugs." ... that is referring to all three FDA-approved drugs? The next sentence following that that talks motor function in type II/III is that across all three drugs or only Risdiplam. Please clarify this better if referring only to a single drug or be exact as to which drug is being discussed.
3. Last sentence same paragraph missing the word. "may" ... "modifying agent (may) not..."
4. the sentence doesn't read well, perhaps place (Jackson Lab; Stock...) after mouse model.
5. is there a reference for this portion of the statement. If so, it would be an excellent reference to include, or if it based on personal evidence that should be noted. "CMV enhancer/chicken- β -actin promoter used to drive SMN in AAV-9-based interventions may not be consistently activated."

Results and Discussion

6. statement of weight for 5058 hets and SMA mice treated with NVS-SM2-treated severe SMA mice weighed ~20 g by PND 30, ... it actually looks like the controls are ~20g and SMA mice are 15g, if it is indeed a 5g difference please simply note this for the SMA mice.

Figures (as ordered in text writing)

7. Supplemental fig. 1. Please correct the nomenclature for FVB/N mice in Fig. 1B and the supplemental text legend. Also, please correct the nomenclature for murine Smn, and EC50 in the legend. It would be helpful to the reader to include the dose, route and length of time of dosing (P2-P7) of mice used in Fig. 1B. Essentially noting what was provided in the results and discussion.
8. Figure 1, it would be advisable to increase the font of the treatment groups above the wells in A-C. At the current size it is very difficult to read.
9. supplemental figure 2. Please indicate what level of significance the various *, ** and *** asterisks indicate.
10. Figure 2 legend should note how many mice were in the vehicle group.

Material and methods:

11. Please reference WO2014028459 as a patent for the readers. Also briefly detail exactly how the NVS-SM2 was prepared for cell culture use and for in vivo dosing. Eg. Was it PEG400:PBS for in

vivo dosing did that require sonication, etc. or readily dissolved.

12. treatment group E dosing. Please better describe what was the 4 times a week dosing schedule (was that 4 straight days a week with a 3-day holiday? This could be important for others trying to replicate this study and the 3hr half-life of NVS-SM2)

13. please provide the reference for the tail genotyping of delta7 SMA mice as noted as "...previously described." The reference is missing.

Reviewer #2 (Comments to the Authors (Required)):

Reitz et al do show some interesting results the key point that the authors do show is that dosing with NVS-SM2 just for the first 3 days in both the delta 7 and Taiwanese's model results in increased survival. What is less clear is the specific affects on the motor neuron. What should be considered as a critical model of SMA of any type is the motor neuron phenotypes. So, the key question I ask the authors is where does this new model relate to the human phenotype and where does it differ? I feel this is not adequately addressed and at least the discussion should be more balanced as to where the mouse may not represent the human condition. After all it will be used to translate therapies to humans. The specific points are as follows.

1) A feature of SMA in humans is the marked impact on motor neurons with minimal to no involvement of other organs (apart from type0). This does not completely translate in the mouse and can cause issues. The fact that this is the case can be determined to a certain extent by looking at the data from the preclinical and clinical use of therapies developed to date. So, when delivered to the Taiwanese SMA mouse via CSF the ASO does not have a large impact but when given subcutaneously it has a much larger impact. In the case of the mouse the BBB is open but in man that will not be the case. So, when you restore SMN in man using intrathecally (CSF) delivered ASO in apparent pre-symptomatic cases (1mV CMAP is below the normal range see) in 3 and 2 copy SMN2 cases there is a remarkable improvement with children gaining the ability to walk and showing no major affects in the periphery too date. (De-Vivo et al. Neuromuscular Disorders Volume 29, Issue 11, November 2019, Pages 842-856) In 2 copy cases the majority did gain the ability to walk but not surprisingly there are more severe children with earlier onset in this group. The response in pre-symptomatic gene therapy trial reported in abstract form is also impressive. So, in an equivalent situation i.e. early in mouse and human the Taiwanese mouse does not have as nearly as impressive response as in human patients most likely because the human condition is mainly driven from the motor neuron while the mouse has additional peripheral issues. The pre-symptomatic treatment compares with treatment in either type 1, type 2 and type 3 once they are symptomatic which is less effective. In essence the same happens in the Delta 7 mice but the motor neuron phenotype is much less pronounced. In the Taiwanese mouse the denervation phenotype is much less evident than Delta7 as indicated in Lin TL, Chen TH, Hsu YY, Cheng YH, Juang BT, Jong YJ. PLoS One. 2016 Apr 28;11(4):e0154723. So what is the significance of peripheral treatments influencing survival in mice has to be considered. I do not fell this is adequately addressed in particular all the discussion and comments interpret the other phenotypes heart, spleen and tail length as directly related to human SMA but really this is not the case in particular for mild SMA cases. The extent of motor behavior analysis is the beam balance which can give an indication of motor correction but really is not fundamental to the motor neuron and can be caused by other things. This should a least be indicated in the discussion or even better some indication of correction of innervation ie in the muscles indicated to be denervated above or using

electrophysiological read outs. This is really to relate it to the human situation so if the authors so choice they can discuss it in this light with the denervation studies to be done latter. Also, instead of classing all phenotype like short tail and small spleen as part of the SMA phenotype I would indicate that these do occur in the mouse but how they correlate to human SMA is unknown. For instance, the autopsy spleen samples reported in the cited paper in some cases show no changes and in other cases very mild changes that are not equivalent to the mouse and likely due to the fact that the patient had an infection in no case is the spleen smaller. Currently there are quite a few so called mild models like the C/C mouse but completely unclear whether these really represent the human phenotype so context is important for the reader.

2) It is quite a surprise that administration at day 8 gives mice that have relatively long survival. The authors state this is not the case for ASOs. However, I would differ when the survival curve for late treatment is examined in the Hua et al paper in figure 1d it is clear that the late treatment results in mice going above 100days and the number of late treated animals seems small (I agree that the figure is not incredibly clear). So, it seems quite likely that late treatment here also could result in two groups of animals there is also limited dosing time for the ASO so the possibility exists that the late treatment effect is a consequence of the animal model used i.e. the Taiwanese mouse as opposed to the delta 7 which has not shown rescue at this late stage. The question is what does this mean clearly in delta 7 SMA mice using inducible transgenes indicates that late SMN restoration is not effective. The question that needs to be approached with caution is where the mice models actually model human SMA and this is not entirely clear. Is it not the case that latter treatment in humans is not as effective? In addition either by electrophysiology (CMAP and MUNE) or by imaging Stam et al. Neuroimage Clin 2019;24:102002. And Smith et al. Clin Imaging Jan-Feb 2019;53:134-137 there appears to be loss of motor neurons in human SMA even in mild cases. Given the simplest interpretation is that motor neurons are lost as SMA progresses with time I can see treatment correction of those motor neurons remaining and maybe an increased CMAP due to sprouting but it seems very unlikely that you can get marked recovery latter in the disease as motor neurons are lost. So what do the authors feel is being recovered with this late treatment and how is it relevant to the overall SMA phenotypes in humans? I feel a more conservative wording is needed in this section, I am not sure what the recovery of a more peripheral phenotype will mean in humans. Also as described below this really differs from the Delta 7 mouse so what models the human situation and why?

3) The late correction at P32 in the paper of Feng et al actually only occurs to the weight of the animals and the overall cross sectional area of the muscle but as stated by Feng et al. innervation changes were not significant. (survival increase is not significant). The following is stated in the Feng et al paper "Switching from the low to the high dose slightly increased the percentage of fully innervated NMJs, although the difference did not reach statistical significance ($\Delta 7$ low-high: 92.3 {plus minus} 4.1%), indicating that switching to high-dose treatment in adulthood may still promote NMJ innervation even though denervation is occurring." Given that it is not significant I think this is a very positive spin on it more like there is no significant improvement in the NMJ. " And at PND60, however, we did not detect any change in the number of L4 ventral root axons when the treatment was switched from low to high dose at PND32 when compared with continuous low-dose treatment ($\Delta 7$ low dose: 753.4 {plus minus} 19.2; $\Delta 7$ low-high: 741 {plus minus} 18; $\Delta 7$ high dose: 803 {plus minus} 25; non-SMA: 910 {plus minus} 13; Fig. 6D). " Taken together, an increased dose of SMN-C3 in the low-dose-treated adult $\Delta 7$ mice resulted in the restoration of the central synapses. This is consistent with human studies and other preclinical work measuring in particular the function of the neuromuscular innervation all be it my different methods. So it makes sense that you could get some rescue of the motor neurons that remain in a patient as those will have restored SMN and be more healthy so can re-sprout and innervate a larger territory which will lead to increase in CMAP

and likely larger muscle fibers. However those neurons lost can not be replaced. It seems likely that the late rescue observed in the Tainwanes mouse is likely not due to recovery of motor neurons but to other phenotypes in the mouse that might or might not be related to SMA thus the discussion of this point needs to be given in context and should be more conservative. Again SMA in humans in particular milder forms of the disease is primarily a motor neuron disease.

4) The others state "optimized low-dose 30-day treatment regimen represents a tractable mild SMA mouse model that resembles the phenotypic delay in human Type II/III SMA patients. Is this really true if the mouse does not have a motor neuron deficient this needs to be toned down it might be a mild model but it is unsure without measures of motor neuron function.

Minor comments

While I know it is often stated that and the authors state the following " duplicated and inverted, resulting in the nearly identical SMN2" The only issue is the SMN2 gene is not always inverted this was the case in the cell line used to generate the YAC libraries that allowed the isolation of the gene. However, the arrangement is not always this way first it is clear that de-novo deletions can arise through homologous recombination see Wirth et al Am. J. Hum. Genet. 61:1102-1111, 1997 second a recent construction of the SMA region clearly shows that the region is not always inverted see Ruhno et al Hum Genet 2019 Mar;138(3):241-256. I think it would be easier to just say it is duplicated without specifics of orientation.

The authors state the following "required for optimal outcomes, and even continuous treatment maybe insufficient to restore full motor function " I feel this statement is misleading and there is simply no evidence for it please modify

The authors state "In het control cohorts, we detected higher levels of human SMN in all tissues compared to severe SMA (Supplemental Figure 2B). We hypothesize that the mouse and human SMN proteins expressed in the het control mice are stabilized due to the oligomerization properties of SMN (Lorson et al, 1998)." I think what you mean here is that the mouse SMN stabilizes the turnover of the human SMN which I would agree with. There is also evidence of feedback loop which alters the degree of SMN exon 7 incorporation is affected by SMN levels. This might be worth mentioning see Ruggiu et al. Mol Cell Biol. 2012 Jan; 32(1): 126-138. and Jodelka et al. Hum. Mol. Genet. 19:4906-4917.

The authors state "revealing tail length as a useful and early phenotypic marker of rescue" Well maybe it really is a pseudo marker as motor neuron improvement may or may not track with it.

The authors state "A drawback of SMN Δ 7 mice is their unfavorable breeding scheme with only 25% of a litter having the SMA genotype." Yes this is true but the question is which mouse model better represents the SMA phenotype and which gives better translation. If it is easier to breed but does not give the same degree of motor neuron involvement I think it is debatable that it is a better model. Can this sentence be modified there are always advantages and disadvantages to different mouse models.

Reviewer #3 (Comments to the Authors (Required)):

Spinal muscular atrophy is a progressive neurological disease resulting from suboptimal levels of the survival motor neuron (SMN) protein. Rodents models of SMA have played a major role in helping identify treatments for SMA. There are now three FDA approved therapies for SMA which has

improved the clinical outcome but patients remain debilitated. The new SMA disease landscape requires new models of the disease. In this manuscript Rietz et al describe a method to generate intermediate models of SMA that more faithfully replicate current patients.

This is a well written study with very detailed experimental information. The methodology is easy to follow and provides a nice new tool in develop new treatment options of SMA.

Minor concerns:

1. The western blots in figure 1 should be quantified.
2. Bar graphs should display individual data points.
3. The statistical tests used in each figure should be included in the figure legends with symbols denoting where significant (as done in supplemental figure 2).

Reviewer #1 (Comments to the Authors (Required)):

This manuscript by Rietz et al. explores both the utility of generating SMA mice of varying disease for future co-therapy studies and extend in vivo efficacy of a SMN2 splice modulator. The manuscript is well written, appropriately detailed and fills a gap the development of drug-induced milder SMA mice that have prolonged survival that are not extremely cumbersome to develop. The comments provided are very minor and provided to enhance details for the reader and replication, extension studies that further explore this model and compound for developing milder SMA mice.

Introduction

1. Please update the following sentence to note the recent FDA approval of RisdiplamTM. "SMN2 splicing modifiers Risdiplam{trade mark, serif} and Branaplam{trade mark, serif} are in Phase 3 clinical trials for SMA type I (NCT02913482) and II (NCT02913482) and Phase 2 for type I (NCT02268552), respectively." **The introduction has been updated and the text has been modified accordingly.**
2. While describing the review by Dangouloff & Servais (2019) of clinical trial results it notes "...from the drugs." ... that is referring to all three FDA-approved drugs? The next sentence following that that talks motor function in type II/III is that across all three drugs or only Risdiplam. Please clarify this better if referring only to a single drug or be exact as to which drug is being discussed. **We agree and have clarified in the text.**
3. Last sentence same paragraph missing the word. "may" ... "modifying agent (may) not..." **Fixed.**
4. the sentence doesn't read well, perhaps place (Jackson Lab; Stock...) after mouse model. **We have removed the stock number as it is also named in the method section.**
5. is there a reference for this portion of the statement. If so, it would be an excellent reference to include, or if it based on personal evidence that should be noted. "CMV enhancer/chicken- β -actin promoter used to drive SMN in AAV-9-based interventions may not be consistently activated." **References have been added: PMID 32576975 and 26942208.**

Results and Discussion

6. statement of weight for 5058 hets and SMA mice treated with NVS-SM2-treated severe SMA mice weighed ~20 g by PND 30, ... it actually looks like the controls are ~20g and SMA mice are 15g, if it is indeed a 5g difference please simply note this for the SMA mice. **Thank you for spotting this. We have updated the text to state both weights, 20g and 15g, at PND30.**

Figures (as ordered in text writing)

7. Supplemental fig. 1. Please correct the nomenclature for FVB/N mice in Fig. 1B and the supplemental text legend. Also, please correct the nomenclature for murine Smn, and EC50 in the legend. It would be helpful to the reader to include the dose, route and length of time of dosing (P2-P7) of mice used in Fig. 1B. Essentially noting what was provided in the results and discussion. **Supplemental Figure 1B has been corrected for nomenclature. The text and legends refer to mouse and human SMN proteins as**

stated in the legend. Mice included in Supp. Fig 1b were left untreated to confirm antibody specificity. The text has been revised to indicate this. We also updated Supplemental Figures 2 and 3 legends according to these recommendations.

8. Figure 1, it would be advisable to increase the font of the treatment groups above the wells in A-C. At the current size it is very difficult to read. **We increased the font size in the table.**

9. Supplemental figure 2. Please indicate what level of significance the various *, ** and *** asterisks indicate. **Added to the figure legend.**

10. Figure 2 legend should note how many mice were in the vehicle group. **Added.**

Material and methods:

11. Please reference WO2014028459 as a patent for the readers. Also briefly detail exactly how the NVS-SM2 was prepared for cell culture use and for in vivo dosing. Eg. Was it PEG400:PBS for in vivo dosing did that require sonication, etc. or readily dissolved. **Patent is now referenced. Details for NVS-SM2 preparation were added to this section.**

12. treatment group E dosing. Please better describe what was the 4 times a week dosing schedule (was that 4 straight days a week with a 3-day holiday? This could be important for others trying to replicate this study and the 3hr half-life of NVS-SM2). **This is explained in the Methods section.**

13. please provide the reference for the tail genotyping of delta7 SMA mice as noted as "...previously described." The reference is missing. **The missing reference has been added.**

Reviewer #2 (Comments to the Authors (Required)):

Reitz et al do show some interesting results the key point that the authors do show is that dosing with NVS-SM2 just for the first 3 days in both the delta 7 and Taiwanese's model results in increased survival. What is less clear is the specific affects on the motor neuron. What should be considered as a critical model of SMA of any type is the motor neuron phenotypes. So, the key question I ask the authors is where does this new model relate to the human phenotype and where does it differ? I feel this is not adequately addressed and at least the discussion should be more balanced as to where the mouse may not represent the human condition. After all it will be used to translate therapies to humans. The specific points are as follows.

We understand and accept the reviewer's points, however there is a fundamental difference in perspective. While transgenic "SMA" mouse models reproduce the SMA genotype, this is limited to the SMN gene and not human SMA and of course engineered mouse SMA is not identical to human SMA. In no way do we wish to infer that our findings are a pretext for human trials. Rather, our view is that this SMN splicing compound works great in two mouse SMA models and therefore can be used to perform experiments that cannot be done in human SMA, including the motor neuron studies the reviewer alludes to. We can totally abrogate disease development or alternatively stop drug and investigate pathogenesis in older mice. That is the primary importance of our findings. While we recognize that this might be viewed as a precursor to a new drug for human SMA therapy that is not our intentions and would be a misinterpretation of our data. We have added text to the discussion to emphasize these points.

We will attempt to respond to some of the detailed comments below from this overarching perspective.

1) A feature of SMA in humans is the marked impact on motor neurons with minimal to no involvement of other organs (apart from type0). This does not completely translate in the mouse and can cause issues. The fact that this is the case can be determined to a certain extent by looking at the data from the preclinical and clinical use of therapies developed to date. So, when delivered to the Taiwanese SMA mouse via CSF the ASO does not have a large impact but when given subcutaneously it has a much larger impact. In the case of the mouse the BBB is open but in man that will not be the case. So, when you restore SMN in man using intrathecally (CSF) delivered ASO in apparent pre-symptomatic cases (1mV CMAP is below the normal range see) in 3 and 2 copy SMN2 cases there is a remarkable improvement with children gaining the ability to walk and showing no major affects in the periphery too date. (De-Vivo et al. *Neuromuscular Disorders* Volume 29, Issue 11, November 2019, Pages 842-856) In 2 copy cases the majority did gain the ability to walk but not surprisingly there are more severe children with earlier onset in this group. The response in pre-symptomatic gene therapy trial reported in abstract form is also impressive. So, in an equivalent situation i.e. early in mouse and human the Taiwanese mouse does not have as nearly as impressive response as in human patients most likely because the human condition is mainly driven from the motor neuron while the mouse has additional peripheral issues. The pre-symptomatic treatment compares with treatment in either type 1, type 2 and type 3 once they are symptomatic which is less effective. In essence the same happens in the Delta 7 mice but the motor neuron phenotype is much less pronounced. In the Taiwanese mouse the denervation phenotype is much less evident than Delta7 as indicated in Lin TL, Chen TH, Hsu YY, Cheng YH, Juang BT, Jong YJ. *PLoS One*. 2016 Apr 28;11(4):e0154723. So what is the significance of peripheral treatments influencing survival in mice has to be considered. I do not feel this is adequately addressed in particular all the discussion and comments interpret the other phenotypes heart, spleen and tail length as directly related to human SMA but really this is not the case in particular for mild SMA cases. The extent of motor behavior analysis is the beam balance which can give an indication of motor correction but really is not fundamental to the motor neuron and can be caused by other things. This should at least be indicated in the discussion or even better some indication of correction of innervation ie in the muscles indicated to be denervated above or using electrophysiological read outs. This is really to relate it to the human situation so if the authors so choose they can discuss it in this light with the denervation studies to be done latter. Also, instead of classing all phenotype like short tail and small spleen as part of the SMA phenotype I would indicate that these do occur in the mouse but how they correlate to human SMA is unknown. For instance, the autopsy spleen samples reported in the cited paper in some cases show no changes and in other cases very mild changes that are not equivalent to the mouse and likely due to the fact that the patient had an infection in no case is the spleen smaller. Currently there are quite a few so called mild models like the C/C mouse but completely unclear whether these really represent the human phenotype so context is important for the reader.

The above discussion is correct, however it deals with treatment of human SMA, which is *not* the purpose of our manuscript. The following is in response to the reviewer's points but is not relevant to the mouse model. Our discoveries can be exploited for detailed investigations of pathogenesis in the murine SMA genotype. We agree that SMA mice do not accurately model human SMA. We demonstrate in this study that with the same limited treatment scheme with NVS-SM2, similar results were obtained in the Taiwanese SMA and the $\Delta 7$ SMA mice. Both mouse models have been reported to have motor neuron defects of varying degrees. Denervation in the $\Delta 7$ SMA mice is more pronounced at end-stage than in the Taiwanese mice. (Lin *et al.*, 2016; Ling *et al.*, 2012; Powis & Gillingwater, 2016). In $\Delta 7$ SMA mice, SMN overexpression is limited to motor neurons (McGovern *et*

al, 2015) or perhaps the majority of neurons using a synapsin promoter (Besse *et al, 2020*) and these only partially rescue, suggesting a degree of dependency on non-neuronal SMN expression. Dependence on SMN in the periphery was also reported with antisense oligonucleotide induction of SMN in the Taiwanese SMA model (Hua *et al, 2011*). However, in *humans* the systemic impact of low SMN proteins is less well established. The accepted disease driver in human SMA is motor neuron death, nevertheless there are reports that SMA patients' peripheral organs are also impacted, such as heart, vasculature, muscle, pancreas and liver (Hensel *et al, 2020*; Lipnick *et al, 2019*). We have modified the discussion to include aspects of the above.

2) It is quite a surprise that administration at day 8 gives mice that have relatively long survival. The authors state this is not the case for ASOs. However, I would differ when the survival curve for late treatment is examined in the Hua *et al* paper in figure 1d it is clear that the late treatment results in mice going above 100days and the number of late treated animals seems small (I agree that the figure is not incredibly clear). So, it seems quite likely that late treatment here also could result in two groups of animals there is also limited dosing time for the ASO so the possibility exists that the late treatment effect is a consequence of the animal model used i.e. the Taiwanese mouse as opposed to the delta 7 which has not shown rescue at this late stage. The question is what does this mean clearly in delta 7 SMA mice using inducible transgenes indicates that late SMN restoration is not effective. The question that needs to be approached with caution is where the mice models actually model human SMA and this is not entirely clear. Is it not the case that latter treatment in humans is not as effective? In addition either by electrophysiology (CMAP and MUNE) or by imaging Stam *et al. Neuroimage Clin 2019;24:102002*. And Smith *et al. Clin Imaging Jan-Feb 2019;53:134-137* there appears to be loss of motor neurons in human SMA even in mild cases. Given the simplest interpretation is that motor neurons are lost as SMA progresses with time I can see treatment correction of those motor neurons remaining and maybe an increased CMAP due to sprouting but it seems very unlikely that you can get marked recovery latter in the disease as motor neurons are lost. So what do the authors feel is being recovered with this late treatment and how is it relevant to the overall SMA phenotypes in humans? I feel a more conservative wording is needed in this section, I am not sure what the recovery of a more peripheral phenotype will mean in humans. Also as described below this really differs from the Delta 7 mouse so what models the human situation and why?

We too were surprised that administration as late as day 8 yielded mice with long term survival to day 110. Few post-symptomatic rescue experiments have been conducted in severe 5058 SMA mice. The study by Hua *et al.* used a late treatment protocol with ASO treatments on PND5 and a second on PND7, which resulted only in a modest increase of survival to PND 16 and a few animals lived past PND 100 (Hua *et al, 2011*). However, when we treated the severe 5058 SMA mice starting PND6, we observed that all severe SMA mice survived till endpoint. We clarified these points in the Discussion.

We agree it would be of scientific interest to investigate whether NVS-SM2 also achieves the same potency in the $\Delta 7$ SMN SMA mice when given in late-stage disease. It is our hope we or others conduct these studies in the future.

3) The late correction at P32 in the paper of Feng *et al* actually only occurs to the weight of the animals and the overall cross sectional area of the muscle but as stated by Feng *et al.* innervation changes were not significant. (survival increase is not significant). The following is stated in the Feng *et al* paper "Switching from the low to the high dose slightly increased the percentage of fully innervated NMJs, although the difference did not reach statistical significance ($\Delta 7$ low-high: 92.3 {plus minus} 4.1%),

indicating that switching to high-dose treatment in adulthood may still promote NMJ innervation even though denervation is occurring." Given that it is not significant I think this is a very positive spin on it more like there is no significant improvement in the NMJ. " And at PND60, however, we did not detect any change in the number of L4 ventral root axons when the treatment was switched from low to high dose at PND32 when compared with continuous low-dose treatment ($\Delta 7$ low dose: 753.4 {plus minus} 19.2; $\Delta 7$ low-high: 741 {plus minus} 18; $\Delta 7$ high dose: 803 {plus minus} 25; non-SMA: 910 {plus minus} 13; Fig. 6D). " Taken together, an increased dose of SMN-C3 in the low-dose-treated adult $\Delta 7$ mice resulted in the restoration of the central synapses. This is consistent with human studies and other preclinical work measuring in particular the function of the neuromuscular innervation all be it my different methods. So it makes sense that you could get some rescue of the motor neurons that remain in a patient as those will have restored SMN and be more healthy so can re-sprout and innervate a larger territory which will lead to increase in CMAP and likely larger muscle fibers. However those neurons lost can not be replaced. It seems likely that the late rescue observed in the Tainwanes mouse is likely not due to recovery of motor neurons but to other phenotypes in the mouse that might or might not be related to SMA thus the discussion of this point needs to be given in context and should be more conservative. Again SMA in humans in particular milder forms of the disease is primarily a motor neuron disease.

Future studies, enabled by our results in this manuscript, will be needed to address these points.

4) The others state "optimized low-dose 30-day treatment regimen represents a tractable mild SMA mouse model that resembles the phenotypic delay in human Type II/III SMA patients. Is this really true if the mouse does not have a motor neuron deficient this needs to be toned down it might be a mild model but it is unsure without measures of motor neuron function. **We agree with the reviewer and have modified the sentence accordingly: This optimized low-dose 30-day treatment regimen may represent a tractable mild SMA mouse model that could resemble the phenotypic delay in human Type II/III SMA patients.**

Minor

comments

While I know it is often stated that and the authors state the following " duplicated and inverted, resulting in the nearly identical SMN2" The only issue is the SMN2 gene is not always inverted this was the case in the cell line used to generate the YAC libraries that allowed the isolation of the gene. However, the arrangement is not always this way first it is clear that de-novo deletions can arise through homologous recombination see Wirth et al Am. J. Hum. Genet. 61:1102-1111, 1997 second a recent construction of the SMA region clearly shows that the region is not always inverted see Ruhno et al Hum Genet 2019 Mar;138(3):241-256. I think it would be easier to just say it is duplicated without specifics of orientation. **We agree with the reviewer and have deleted the word duplicated.**

The authors state the following "required for optimal outcomes, and even continuous treatment maybe insufficient to restore full motor function" I feel this statement is misleading and there is simply no evidence for it please modify. **We agree with the reviewer and have deleted the sentence.**

The authors state "In het control cohorts, we detected higher levels of human SMN in all tissues compared to severe SMA (Supplemental Figure 2B). We hypothesize that the mouse and human SMN proteins expressed in the het control mice are stabilized due to the oligomerization properties of SMN (Lorson et al, 1998)." I think what you mean here is that the mouse SMN stabilizes the turnover of the

human SMN which I would agree with. There is also evidence of feedback loop which alters the degree of SMN exon 7 incorporation is affected by SMN levels. This might be worth mentioning see Ruggiu et al. Mol Cell Biol. 2012 Jan; 32(1): 126-138. and Jodelka et al. Hum. Mol. Genet. 19:4906-4917. **We have modified the sentence to now read: We hypothesize that the human SMN proteins expressed in the het control mice are increased by mouse SMN proteins due to the oligomerization properties of SMN (Lorson *et al*, 1998) and/or through increased SMN exon 7 incorporation due to higher SMN protein levels (Jodelka *et al*, 2010; Ruggiu *et al*, 2012).**

The authors state "revealing tail length as a useful and early phenotypic marker of rescue" Well maybe it really is a pseudo marker as motor neuron improvement may or may not track with it. **Although that there is no study that investigated whether tail length tracks motor neuron improvement, several studies reported that treatment or genetic modulation resulted in improvement of motor function and normalized development of the tail.**

The authors state "A drawback of SMN Δ 7 mice is their unfavorable breeding scheme with only 25% of a litter having the SMA genotype." Yes this is true but the question is which mouse model better represents the SMA phenotype and which gives better translation. If it is easier to breed but does not give the same degree of motor neuron involvement I think it is debatable that it is a better model. Can this sentence be modified there are always advantages and disadvantages to different mouse models.

We have modified the sentence : The SMN Δ 7 mouse breeding scheme produces a predicted 25% litter with the SMA genotype.

Reviewer #3 (Comments to the Authors (Required)):

Spinal muscular atrophy is a progressive neurological disease resulting from suboptimal levels of the survival motor neuron (SMN) protein. Rodents models of SMA have played a major role in helping identify treatments for SMA. There are now three FDA approved therapies for SMA which has improved the clinical outcome but patients remain debilitated. The new SMA disease landscape requires new models of the disease. In this manuscript Rietz et al describe a method to generate intermediate models of SMA that more faithfully replicate current patients.

This is a well written study with very detailed experimental information. The methodology is easy to follow and provides a nice new tool in develop new treatment options of SMA.

Minor concerns:

1. The western blots in figure 1 should be quantified. **Quantifications are presented in Supplemental Figure 2.**
2. Bar graphs should display individual data points. **All bar graphs have been updated accordingly.**
3. The statistical tests used in each figure should be included in the figure legends with symbols denoting where significant (as done in supplemental figure 2). **The p-values have been added to respective figure legends.**

November 2, 2020

RE: Life Science Alliance Manuscript #LSA-2020-00889-TR

Dr. Elliot J Androphy
Indiana University School of Medicine
Department of Dermatology 545 Barnhill Dr. Emerson Hall 139
Emerson 138
Indianapolis, IN 46202

Dear Dr. Androphy,

Thank you for submitting your revised manuscript entitled "Short duration splice promoting compound enables a tunable mouse model of spinal muscular atrophy". We would be happy to publish your paper in Life Science Alliance pending final revisions necessary to meet our formatting guidelines.

Along with the points listed below, please also attend to the following:

- please consult our Manuscript Preparation Guidelines <https://www.life-science-alliance.org/manuscript-prep> and put your manuscript sections in the correct order
- please separate the Results and discussion into 2 separate sections - one results section, and one discussion section
- please add the Author Contributions to the main manuscript text
- please add a Running Title to our system
- please add a conflict of interest statement to your main manuscript text
- please double-check your main figure legends to add a legend for Figure 8; and add your supplementary figure and video legends to your main manuscript text, directly under the main figure legends
- please add a callout for Supplemental Figure 3B to your main manuscript text

A. FINAL FILES:

B. MANUSCRIPT ORGANIZATION AND FORMATTING:

Sincerely,

Shachi Bhatt, Ph.D.
Executive Editor
Life Science Alliance

<https://www.lsjournal.org/>
Tweet @SciBhatt @LSAJournal

November 9, 2020

RE: Life Science Alliance Manuscript #LSA-2020-00889-TRR

Dr. Elliot J Androphy
Indiana University School of Medicine
Department of Dermatology 545 Barnhill Dr. Emerson Hall 139
Emerson 138
Indianapolis, IN 46202

Dear Dr. Androphy,

Thank you for submitting your Research Article entitled "Short duration splice promoting compound enables a tunable mouse model of spinal muscular atrophy". It is a pleasure to let you know that your manuscript is now accepted for publication in Life Science Alliance. Congratulations on this interesting work.

DISTRIBUTION OF MATERIALS:

Again, congratulations on a very nice paper. I hope you found the review process to be constructive and are pleased with how the manuscript was handled editorially. We look forward to future exciting submissions from your lab.

Sincerely,

Shachi Bhatt, Ph.D.

Executive Editor

Life Science Alliance

<https://www.lsjournal.org/>
